# Reconciling meta-learning and continual learning with online mixtures of tasks

**Ghassen Jerfel** [*]
gj47@duke.edu
Duke University

**Erin Grant** [*]
eringrant@berkeley.edu
UC Berkeley

**Thomas L. Griffiths**
tomg@princeton.edu
Princeton University

**Katherine Heller**
kheller@stat.duke.edu
Duke University

## Abstract

*Learning-to-learn* or *meta-learning* leverages data-driven inductive bias to increase the efficiency of learning on a novel task. This approach encounters difficulty when transfer is not advantageous, for instance, when tasks are considerably dissimilar or change over time. We use the connection between gradient-based meta-learning and hierarchical Bayes to propose a Dirichlet process mixture of hierarchical Bayesian models over the parameters of an arbitrary parametric model such as a neural network. In contrast to consolidating inductive biases into a single set of hyperparameters, our approach of task-dependent hyperparameter selection better handles latent distribution shift, as demonstrated on a set of evolving, image-based, few-shot learning benchmarks.

## 1 Introduction

Meta-learning algorithms aim to increase the efficiency of learning by treating task-specific learning episodes as examples from which to generalize [47]. The central assumption of a meta-learning algorithm is that some tasks are inherently related and so inductive transfer can improve sample efficiency and generalization [8, 9, 5]. In learning a single set of domain-general hyperparameters that parameterize a metric space [53] or an optimizer [40, 14], recent meta-learning algorithms make the assumption that tasks are equally related, and therefore non-adaptive, mutual transfer is appropriate. This assumption has been cemented in recent few-shot learning benchmarks, which comprise a set of tasks generated in a uniform manner [*e.g.,* 53, 14].

However, the real world often presents scenarios in which an agent must decide what degree of transfer is appropriate. In some cases, a subset of tasks are more strongly related to each other, and so non-uniform transfer provides a strategic advantage. On the other hand, transfer in the presence of dissimilar or outlier tasks worsens generalization performance [44, 12]. Moreover, when the underlying task distribution is non-stationary, inductive transfer to previously observed tasks should exhibit graceful degradation to address the catastrophic forgetting problem [28]. In these settings, the consolidation of all inductive biases into a single set of hyperparameters is not well-posed to deal with changing or diverse tasks. In contrast, in order to account for this degree of task heterogeneity, humans detect and adapt to novel contexts by attending to relationships between tasks [10].

In this work, we learn a mixture of hierarchical models that allows a meta-learner to adaptively select over a set of learned parameter initializations for gradient-based adaptation to a new task. The method is equivalent to clustering task-specific parameters in the hierarchical model induced by

---

[*]Equal contribution.

recasting gradient-based meta-learning as hierarchical Bayes [21] and generalizes the model-agnostic meta-learning (MAML) algorithm introduced in [14]. By treating the assignment of task-specific parameters to clusters as latent variables, we can directly detect similarities between tasks on the basis of the task-specific likelihood, which may be parameterized by an expressive model such as a neural network. Our approach, therefore, alleviates the need for explicit geometric or probabilistic modeling assumptions about the weights of a complex parametric model and provides a scalable method to regulate information transfer between episodes.

We additionally consider the setting of a non-stationary or evolving task distribution, which necessitates a meta-learning method that possesses adaptive complexity. We translate stochastic point estimation in an infinite mixture [39] over model parameters into a gradient-based meta-learning algorithm that is compatible with any differentiable likelihood model and requires no distributional assumptions. We demonstrate the unexplored ability of non-parametric priors over neural network parameters to automatically detect and adapt to task distribution shift in a naturalistic image dataset; addressing the non-trivial setting of *task-agnostic* continual learning in which the task change is unobserved [*c.f., task-aware* settings such as 28].

## 2 Gradient-based meta-learning as hierarchical Bayes

Since our approach is grounded in the probabilistic formulation of meta-learning as hierarchical Bayes [4], our approach can be applied to any probabilistic meta-learner. In this work, we focus on model-agnostic meta-learning (MAML) [14], a gradient-based meta-learning approach that estimates global parameters to be shared among task-specific models as an initialization for a few steps of gradient descent. MAML admits a natural interpretation as parameter estimation in a hierarchical probabilistic model, where the learned initialization acts as data-driven regularization for the estimation of task-specific parameters $\hat{\phi}_j$.

In particular, [21] cast MAML as posterior inference for task-specific parameters $\phi_j$ given some samples of task-specific data $\boldsymbol{x}_{j_{1:N}}$ and a prior over $\phi_j$ that is induced by the early stopping of an iterative optimization procedure; truncation at $K$ steps of gradient descent on the negative log-likelihood $-\log p(\boldsymbol{x}_{j_{1:N}} \mid \phi_j)$ starting from $\phi_{j_{(0)}} = \boldsymbol{\theta}$ can be then understood as mode estimation of the posterior $p(\phi_j \mid \boldsymbol{x}_{j_{1:N}}, \boldsymbol{\theta})$. The mode estimates $\hat{\phi}_j = \phi_{j_{(0)}} + \alpha \sum_{k=1}^{K} \nabla_\phi \log p(\boldsymbol{x}_{j_{1:N}} \mid \phi_{j_{(k-1)}})$ are then combined to evaluate the marginal likelihood for each task as

$$p\left(\boldsymbol{x}_{j_{N+1:N+M}} \mid \boldsymbol{\theta}\right) = \int p(\boldsymbol{x}_{j_{N+1:N+M}} \mid \phi_j) p(\phi_j \mid \boldsymbol{\theta}) \, \mathrm{d}\phi_j \approx p(\boldsymbol{x}_{j_{N+1:N+M}} \mid \hat{\phi}_j), \tag{1}$$

where $\boldsymbol{x}_{j_{N+1:N+M}}$ is another set of samples from the $j$th task. A training dataset can then be summarized in an empirical Bayes point estimate of $\boldsymbol{\theta}$ computed by gradient-based optimization of the joint marginal likelihood in (1) in across tasks, so that the likelihood of a datapoint sampled from a new task can be computed using only $\boldsymbol{\theta}$ and without storing the task-specific parameters.

## 3 Improving meta-learning by modeling latent task structure

If the task distribution is heterogeneous, assuming a single parameter initialization $\boldsymbol{\theta}$ for gradient-based meta-learning is not suitable because it is unlikely that the point estimate computed by a few steps of gradient descent will sufficiently adapt the task-specific parameters $\phi$ to a diversity of tasks. Moreover, explicitly estimating relatedness between tasks has the potential to aid the efficacy of a meta-learning algorithm by modulating both positive and negative transfer [52, 60, 45, 61], and by identifying outlier tasks that require a more significant degree of adaptation [56, 23]. Nonetheless, defining an appropriate notion of task relatedness is a difficult problem in the high-dimensional parameter or activation space of models such as neural networks.

Using the probabilistic interpretation of Section 2, we deal with the variability in the tasks by assuming that each set of task-specific parameters $\phi_j$ is drawn from a mixture of base distributions, each of which is parameterized by a hyperparameter $\boldsymbol{\theta}^{(\ell)}$. Accordingly, we capture task relatedness by estimating the likelihood of assigning each task to a mixture component based simply on the task-specific negative log-likelihood after a few steps of gradient-based adaptation. The result is a scalable meta-learning algorithm that jointly learns task-specific cluster assignments and model parameters, and is capable of modulating the transfer of information across tasks by clustering together related task-specific parameter settings.

---

**Algorithm 1** `Stochastic gradient-based EM for `finite` and `infinite` mixtures(`
*dataset $\mathscr{D}$, meta-learning rate $\beta$, adaptation rate $\alpha$, temperature $\tau$, initial cluster count $L_0$, meta-batch size $J$, training batch size $N$, validation batch size $M$, adaptation iteration count $K$, global prior $G_0$)*

---

Initialize cluster count $L \leftarrow L_0$ and meta-level parameters $\boldsymbol{\theta}^{(1)}, \dots, \boldsymbol{\theta}^{(L)} \sim G_0$
**while** *not converged* **do**
  Draw tasks $\mathcal{T}_1, \dots, \mathcal{T}_J \sim p_{\mathscr{D}}(\mathcal{T})$
  **for** *j in* $1, \dots, J$ **do**
    Draw task-specific datapoints, $\boldsymbol{x}_{j_1} \dots \boldsymbol{x}_{j_{N+M}} \sim p_{\mathcal{T}_j}(\boldsymbol{x})$
    Draw a parameter initialization for a new cluster from the global prior, $\boldsymbol{\theta}^{(L+1)} \sim G_0$
    **for** $\ell$ *in* $\{1, \dots, L, L+1\}$ **do**
      Initialize $\hat{\boldsymbol{\phi}}_j^{(\ell)} \leftarrow \boldsymbol{\theta}^{(\ell)}$
      Compute task-specific mode estimate, $\hat{\boldsymbol{\phi}}_j^{(\ell)} \leftarrow \hat{\boldsymbol{\phi}}_j^{(\ell)} + \alpha \sum_k \nabla_{\boldsymbol{\phi}} \log p(\boldsymbol{x}_{j_{1:N}} \mid \hat{\boldsymbol{\phi}}_j^{(\ell)})$
    Compute assignment of tasks to clusters, $\gamma_j \leftarrow$ `E-STEP` $(\boldsymbol{x}_{j_{1:N}}, \hat{\boldsymbol{\phi}}_j^{(1:L)})$
  Update each component $\ell$ in $1, \dots, L$, $\boldsymbol{\theta}^{(\ell)} \leftarrow \boldsymbol{\theta}^{(\ell)} +$ `M-STEP` $(\{\boldsymbol{x}_{j_{N+1:N+M}}, \hat{\boldsymbol{\phi}}_j^{(\ell)}, \gamma_j\}_{j=1}^J)$
  Summarize $\{\boldsymbol{\theta}_1, \dots\}$ to update global prior $G_0$
**return** $\{\boldsymbol{\theta}^{(1)}, \dots\}$

---

| `E-STEP(` $\{\boldsymbol{x}_{j_i}\}_{i=1}^N, \{\hat{\boldsymbol{\phi}}_j^{(\ell)}\}_{\ell=1}^L$ `)` | `M-STEP(` $\{\boldsymbol{x}_{j_i}\}_{i=1}^M, \hat{\boldsymbol{\phi}}_j^{(\ell)}, \gamma_j$ `)` |
|---|---|
| **return** $\tau$-$\mathrm{softmax}_\ell(\sum_i \log p(\boldsymbol{x}_{j_i} \mid \hat{\boldsymbol{\phi}}_j^{(\ell)}))$ | **return** $\beta \nabla_{\boldsymbol{\theta}}[\sum_{j,i} \gamma_j \log p(\boldsymbol{x}_{j_i} \mid \hat{\boldsymbol{\phi}}_j^{(\ell)})]$ |

Top: **Algorithm 1:** Stochastic gradient-based expectation maximization (EM) for probabilistic clustering of task-specific parameters in a meta-learning setting. Bottom: **Subroutine 2:** The `E-STEP` and `M-STEP` for a finite mixture of hierarchical Bayesian models implemented as gradient-based meta-learners.

Formally, let $\boldsymbol{z}_j$ be the categorical latent variable indicating the cluster assignment of each task-specific parameter $\boldsymbol{\phi}_j$. Direct maximization of the mixture model likelihood is a combinatorial optimization problem that can grow intractable. This intractability is equally problematic for the posterior distribution over the cluster assignment variables $\boldsymbol{z}_j$ and the task-specific parameters $\boldsymbol{\phi}_j$, which are both treated as latent variables in the probabilistic formulation of meta-learning. A scalable approximation involves representing the conditional distribution for each latent variable with a *maximum a posteriori* (MAP) estimate. In our meta-learning setting of a mixture of hierarchical Bayes (HB) models, this suggests an augmented expectation maximization (EM) procedure [13] alternating between an `E-STEP` that computes an expectation of the task-to-cluster assignments $\boldsymbol{z}_j$, which itself involves the computation of a conditional mode estimate for the task-specific parameters $\boldsymbol{\phi}_j$, and an `M-STEP` that updates the hyperparameters $\theta^{(1:L)}$ (see Subroutine 2).

To ensure scalability, we use the minibatch variant of stochastic optimization [43] in both the `E-STEP` and the `M-STEP`; such approaches to EM are motivated by a view of the algorithm as optimizing a single free energy at both the `E-STEP` and the `M-STEP` [37]. In particular, for each task $j$ and cluster $\ell$, we follow the gradients to minimize the negative log-likelihood on the training data points $\boldsymbol{x}_{j_{1:N}}$, using the cluster parameters $\boldsymbol{\theta}^{(\ell)}$ as initialization. This allows us to obtain a modal point estimate of the task-specific parameters $\hat{\boldsymbol{\phi}}_j^{(\ell)}$. The `E-STEP` in Subroutine 2 leverages the connection between gradient-based meta-learning and HB [21] and the differentiability of our clustering procedure to employ the task-specific parameters to compute the posterior probability of cluster assignment. Accordingly, based on the likelihood of the same training data points under the model parameterized by $\hat{\boldsymbol{\phi}}_j^{(\ell)}$, we compute the cluster assignment probabilities as

$$\gamma_j^{(\ell)} := p(\boldsymbol{z}_j = \ell \mid \boldsymbol{x}_{j_{1:N}}, \boldsymbol{\theta}^{(1:L)}) \propto \int p(\boldsymbol{x}_{j_{1:N}} \mid \boldsymbol{\phi}_j) \, p(\boldsymbol{\phi}_j \mid \boldsymbol{\theta}^{(\ell)}) \, \mathrm{d}\boldsymbol{\phi}_j \approx p(\boldsymbol{x}_{j_{1:N}} \mid \hat{\boldsymbol{\phi}}_j^{(\ell)}). \quad (2)$$

The cluster means $\boldsymbol{\theta}^{(\ell)}$ are then updated by gradient descent on the validation loss in the `M-STEP` in Subroutine 2; this `M-STEP` is analogous to the MAML algorithm in [14] with the addition of mixing weights $\gamma_j^{(\ell)}$.

Note that, unlike other recent approaches to probabilistic clustering [*e.g.,* 3] we adhere to the episodic meta-learning setup for both training and testing since only the task support set $\boldsymbol{x}_{j_{1:N}}$ is used to compute both the point estimate $\hat{\boldsymbol{\phi}}_j^{(\ell)}$ and the cluster responsibilities $\gamma_j^{(\ell)}$. See Algorithm 1 for the full algorithm, whose high-level structure is shared with the non-parametric variant of our method detailed in Section 5.

**Table 1:** Meta-test set accuracy on the *mini*ImageNet **5-way**, **1-** and **5-shot** classification benchmarks from [53] among methods using a comparable architecture (the 4-layer convolutional network from [53]). For methods on which we report results in later experiments, we additionally report the total number of parameters optimized by the meta-learning algorithm. [a] Results reported by [40]. [b] We report test accuracy for models matching train and test "shot" and "way". [c] We report test accuracy for a comparable base (task-specific network) architecture.

| Model | Num. param. | 1-shot (%) | 5-shot (%) |
|---|---|---|---|
| **matching network** [53][a] | | $43.56 \pm 0.84$ | $55.31 \pm 0.73$ |
| **meta-learner LSTM** [40] | | $43.44 \pm 0.77$ | $60.60 \pm 0.71$ |
| **prototypical networks** [49][b] | | $46.61 \pm 0.78$ | $65.77 \pm 0.70$ |
| **MAML** [14] | | $48.70 \pm 1.84$ | $63.11 \pm 0.92$ |
| **MT-net** [30] | $38,907$ | $51.70 \pm 1.84$ | |
| **PLATIPUS** [15] | $65,546$ | $50.13 \pm 1.86$ | |
| **VERSA** [20][c] | $807,938$ | $48.53 \pm 1.84$ | |
| **Our method**: 2 components | $65,546$ | $49.60 \pm 1.50$ | $64.60 \pm 0.92$ |
| 3 components | $98,319$ | $51.20 \pm 1.52$ | $65.00 \pm 0.96$ |
| 4 components | $131,092$ | $50.49 \pm 1.46$ | $64.78 \pm 1.43$ |
| 5 components | $163,865$ | $51.46 \pm 1.68$ | |

## 4    Experiment: *mini*ImageNet few-shot classification

Clustering task-specific parameters provides a way for a meta-learner to deal with task heterogeneity as each cluster can be associated with a subset of the tasks that would benefit most from mutual transfer. While we do not expect existing tasks to present a significant degree of heterogeneity given the uniform sampling assumptions behind their design, we nevertheless conduct an experiment to validate that our method gives an improvement on a standard benchmark for few-shot learning.

We apply Algorithm 3 with Subroutine 2 and $L \in \{2, 3, 4, 5\}$ components to the 1-shot and 5-shot, 5-way, *mini*ImageNet few-shot classification benchmarks [53]; Appendix C.2.1 contains additional experimental details. We demonstrate in Table 1 that a mixture of meta-learners improves the performance of gradient-based meta-learning on this task for any number of components. However, the performance of the parametric mixture does not improve monotonically with the number of components $L$. This leads us to the development of non-parametric clustering for continual meta-learning, where enforcing specialization to subgroups of tasks and increasing model complexity is, in fact, necessary to preserve performance on prior tasks due to significant heterogeneity.

## 5    Scalable online mixtures for task-agnostic continual learning

The mixture of meta-learners developed in Section 3 addresses a drawback of meta-learning approaches such as MAML that consolidate task-general information into a single set of hyperparameters. However, the method adds another dimension to model selection in the form of identifying the correct number of mixture components. While this may be resolved by cross-validation if the dataset is static and therefore the number of components can remain fixed, adhering to a fixed number of components throughout training is not appropriate in the non-stationary regime, where the underlying task distribution changes as different types of tasks are presented sequentially in a continual learning setting. In this regime, it is important to incrementally introduce more components that can each specialize to the distribution of tasks observed at the time of spawning.

To address this, we derive a scalable stochastic estimation procedure to compute the expectation of task-to-cluster assignments (E-STEP) for a growing number of task clusters in a *non-parametric* mixture model [39] called the Dirichlet process mixture model (DPMM). The formulation of the Dirichlet process mixture model (DPMM) that is most appropriate for incremental learning is the sequential draws formulation that corresponds to an instantiation of the Chinese restaurant process (CRP) [39]. A CRP prior over $z_j$ allows some probability to be assigned to a new mixture component while the task identities are inferred in a sequential manner, and has therefore been key to recent online and stochastic learning of the DPMM [31]. A draw from a CRP proceeds as follows: For a sequence of tasks, the first task is assigned to the first cluster and the $j$th subsequent task is then assigned to the $\ell$th cluster with probability

$$p\left(z_j = \ell \mid z_{1:j-1}, \zeta\right) = \begin{cases} n^{(\ell)}/n + \zeta & \text{for } \ell \leq L \\ \zeta/n + \zeta & \text{for } \ell = L+1 , \end{cases} \tag{3}$$

E-STEP( $\boldsymbol{x}_{j_{1:N}}, \hat{\boldsymbol{\phi}}_j^{(1:L)}$, *concentration* $\zeta$, *threshold* $\epsilon$)

    DPMM log-likelihood for all $\ell$ in $1, \ldots, L$, $\rho_j^{(\ell)} \leftarrow \sum_i \log p(\boldsymbol{x}_{j_i} \mid \hat{\boldsymbol{\phi}}_j^{(\ell)}) + \log n^{(\ell)}$

    DPMM log-likelihood for new component, $\rho_j^{(L+1)} \leftarrow \sum_i \log p(\boldsymbol{x}_{j_i} \mid \hat{\boldsymbol{\phi}}_j^{(L+1)}) + \log \zeta$

    DPMM assignments, $\gamma_j \leftarrow \tau\text{-softmax}(\rho_j^{(1)}, \ldots, \rho_j^{(L+1)})$

    **if** $\gamma_j^{(L+1)} > \epsilon$ **then**

        Expand the model by incrementing $L \leftarrow L + 1$

    **else**

        Renormalize $\gamma_j \leftarrow \tau\text{-softmax}(\rho_j^{(1)}, \ldots, \rho_j^{(L)})$

    **return** $\gamma_j$

---

M-STEP( $\{\boldsymbol{x}_{j_i}\}_{i=1}^M, \hat{\boldsymbol{\phi}}_j^{(\ell)}, \gamma_j$, *concentration* $\zeta$)

    **return** $\beta \nabla_{\boldsymbol{\theta}}[\sum_{j,i} \gamma_j \log p(\boldsymbol{x}_{j_i} \mid \hat{\boldsymbol{\phi}}_j^{(\ell)}) + \log n^{(\ell)}]$

**Subroutine 3:** The `E-STEP` and `M-STEP` for an infinite mixture of hierarchical Bayesian models.

where $L$ is the number of non-empty clusters, $n^{(\ell)}$ is the number of tasks already occupying a cluster $\ell$, and $\zeta$ is a fixed positive concentration parameter. The prior probability associated with a new mixture component is therefore $p(\boldsymbol{z}_j = L + 1 \mid \boldsymbol{z}_{1:j-1}, \zeta)$.

In a similar spirit to Section 3, we develop a stochastic EM procedure for the estimation of the latent task-specific parameters $\boldsymbol{\phi}_{1:J}$ and the meta-level parameters $\boldsymbol{\theta}^{(1:L)}$ in the DPMM, which allows the number of observed task clusters to grow in an online manner with the diversity of the task distribution. While computation of the mode estimate of the task-specific parameters $\boldsymbol{\phi}_j$ is mostly unchanged from the finite variant, the estimation of the cluster assignment variables $\boldsymbol{z}$ in the E-STEP requires revisiting the Gibbs conditional distributions due to the potential addition of a new cluster at each step. For a DPMM, the conditional distributions for $\boldsymbol{z}_j$ are

$$p(\boldsymbol{z}_j = \ell \mid \boldsymbol{x}_{j_{1:M}}, \boldsymbol{z}_{1:j-1}) \propto \begin{cases} n^{(\ell)} \int p(\boldsymbol{x}_{j_{1:M}} \mid \boldsymbol{\phi}_j^{(\ell)}) p(\boldsymbol{\phi}_j^{(\ell)} \mid \boldsymbol{\theta}) \, \mathrm{d}\boldsymbol{\phi}_j \, \mathrm{d}G_\ell(\boldsymbol{\theta}) & \text{for } \ell \leq L \\ \zeta \int p(\boldsymbol{x}_{j_{1:M}} \mid \boldsymbol{\phi}_j^{(0)}) p(\boldsymbol{\phi}_j^{(0)} \mid \boldsymbol{\theta}) \, \mathrm{d}\boldsymbol{\phi}_j \, \mathrm{d}G_0(\boldsymbol{\theta}) & \text{for } \ell = L + 1 \end{cases} \quad (4)$$

with $G_0$ as the base measure or global prior over the components of the CRP, $G_\ell$ is the prior over each cluster's parameters, initialized with a draw from a Gaussian centered at $G_0$ with a fixed variance and updated over time using summary statistics from the set of active components $\{\boldsymbol{\theta}^{(0)}, \ldots, \boldsymbol{\theta}^{(L)}\}$.

Taking the logarithm of the posterior over task-to-cluster assignments $\boldsymbol{z}_j$ in (4) and using a mode estimate $\hat{\boldsymbol{\phi}}_j^{(\ell)}$ for task-specific parameters $\boldsymbol{\phi}_j$ as drawn from the $\ell$th cluster gives the E-STEP in Subroutine 3. We may also omit the prior term $\log p(\hat{\boldsymbol{\phi}}_j^{(\ell)} \mid \boldsymbol{\theta}^{(\ell)})$ as it arises as an implicit prior resulting from truncated gradient descent, as explained in Section 3 of [21].

## 6 Experiments: *Task-agnostic* continual few-shot regression & classification

By treating the assignment of tasks to clusters as latent variables, the algorithm of Section 5 can adapt to a changing distribution of tasks, without any external information to signal distribution shift (*i.e.,* in a *task-agnostic* manner). Here, we present our main experimental results on both a novel synthetic regression benchmark as well as a novel evolving variant of *mini*ImageNet, and confirm the algorithm's ability to adapt to distribution shift by spawning a newly specialized cluster.

**High-capacity baselines.** As an ablation, we compare to the **non-uniform** parametric **mixture** proposed in Section 3 with the number of components fixed at the total number of task distributions in the dataset (3). We also consider a **uniform** parametric **mixture** in which each component receives equal assignments; this can also be seen as the non-uniform mixture in the infinite temperature ($\tau$) limit. Note that our meta-learner has a lower capacity than these two baselines for most of the training procedure, as it may decide to expand its capacity past one component only when the task distribution changes. Finally, for the large-scale experiment in Section 6.2, we compare with three recent meta-learning algorithms that report improved performance on the standard *mini*ImageNet benchmark of Section 3, but are not explicitly posed to address the continual learning setting of evolving tasks: **MT-net** [30], **PLATIPUS** [15], and **VERSA** [20].

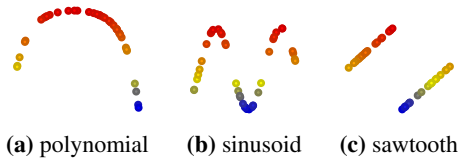

**(a)** polynomial  **(b)** sinusoid  **(c)** sawtooth

**Figure 4:** The diverse set of periodic functions used for few-shot regression in Section 6.1.

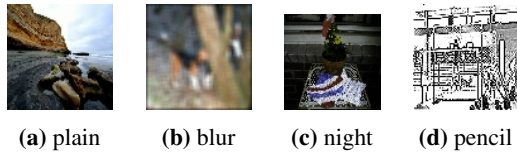

**(a)** plain  **(b)** blur  **(c)** night  **(d)** pencil

**Figure 5:** Artistic filters **(b-d)** applied to *mini*ImageNet **(a)** to ensure non-homogeneity of tasks in Section 6.2.

## 6.1 Continual few-shot regression

We first consider an explanatory experiment in which three regression tasks are presented sequentially with no overlap. For input $x$ sampled uniformly from $[-5, 5]$, each regression task is generated, in a similar spirit to the sinusoidal regression setup in [14], from one of a set of simple but distinct one-dimensional functions (polynomial Figure 4a, sinusoid wave Figure 4b, and sawtooth wave Figure 4c). For the experiment in Figure 6 and Figure 7, we presented the polynomial tasks for 4000 iterations, followed by sinusoid tasks for 3000 iterations, and finally sawtooth tasks. Additional details on the experimental setup can be found in Appendix C.2.2.

**Results: Distribution shift detection.** The cluster responsibilities in Figure 7 on the meta-test dataset of tasks, from each of the three regression types in Figure 4, indicate that the non-parametric algorithm recognizes a change in the task distribution and spawns a new cluster at iteration 4000 and promptly after iteration 7000. Each newly created cluster is specialized to the task distribution observed at the time of spawning and remains as such throughout training, since the majority of assignments for each type of regression remains under a given cluster from the time of its introduction.

**Results: Improved generalization and slower degradation of performance.** We investigate the progression of the meta-test mean-squared error (MSE) for the three regression task distributions in Figure 6. We first note the clear advantage of non-uniform assignment both in improved generalization, when testing on the active task distribution, and in slower degradation, when testing on previous distributions. This is due to the ability of these methods to modulate the transfer of information in order to limit negative transfer. In contrast, the uniform method cannot selectively adapt specific clusters to be responsible for any given task, and thus inevitably suffers from catastrophic forgetting.

The adaptive capacity of our non-parametric method allows it to spawn clusters that specialize to newly observed tasks. Accordingly, even if the overall capacity is lower than that of the comparable non-uniform parametric method, our method achieves similar or better generalization, at any given training iteration. More importantly, specialization allows our method to better modulate information transfer as the clusters are better differentiated. Consequently, each cluster does not account for many assignments from more than a single task distribution throughout training. Therefore, we observed a significantly slower rate of degradation of the MSE on previous task distributions as new tasks are introduced. This is especially evident from the performance on the first task in Figure 6.

## 6.2 Continual few-shot classification

Next, we consider an evolving variant of the *mini*ImageNet few-shot classification task. In this variant, one of a set of artistic filters are applied to the images during the meta-training procedure to simulate a changing distribution of few-shot classification tasks. For the experiment in Figure 8 and Figure 9 we first train using images with a "blur" filter (Figure 5b) for 7500 iterations, then with a "night" filter (Figure 5c) for another 7500 iterations, and finally with a "pencil" filter (Figure 5d). Additional details on the experimental setup can be found in Appendix C.2.3.

**Results: Meta-test accuracy.** In Figure 9, we report the evolution of the meta-test accuracy for two variants of our non-parametric meta-learner in comparison to the parametric baselines introduced in Section 6, *high-capacity baselines*. The *task-agnostic* variant is the core algorithm described in previous sections, as used for the regression tasks. The *task-aware* variant augments the core algorithm with a cool-down period that prevents over-spawning for the duration of a training phase. This requires some knowledge of the duration which is external to the meta-learner, thus the *task-aware* nomenclature (note that this does not correspond to a true oracle, as we do not enforce spawning of a cluster; see Appendix D.1 for further details).

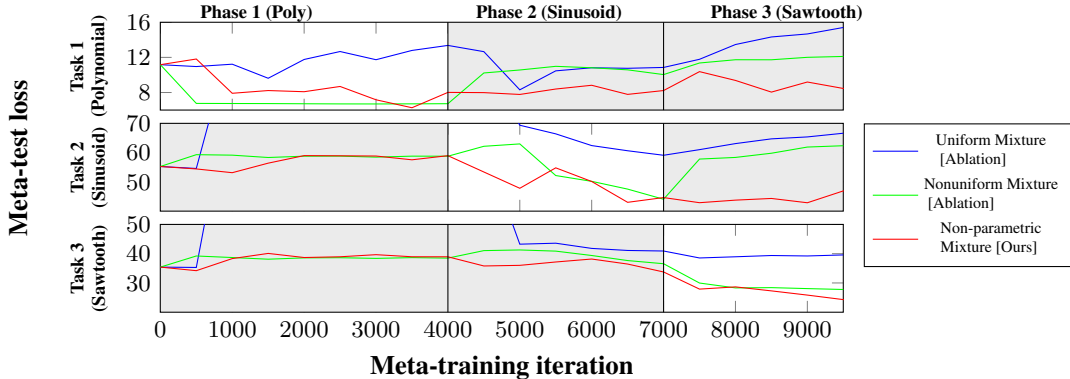

**Figure 6:** Results on the evolving dataset of few-shot regression tasks (lower is better). Each panel (row) presents, for a specific task type (polynomial, sinusoid or sawtooth), the average meta-test set accuracy of each method over cumulative number of few-shot episodes. We additionally report the degree of loss in backward transfer (i.e., catastrophic forgetting) to the tasks in each meta-test set in the legend; all methods but the non-parametric method experience a large degree of catastrophic forgetting during an inactive phase.

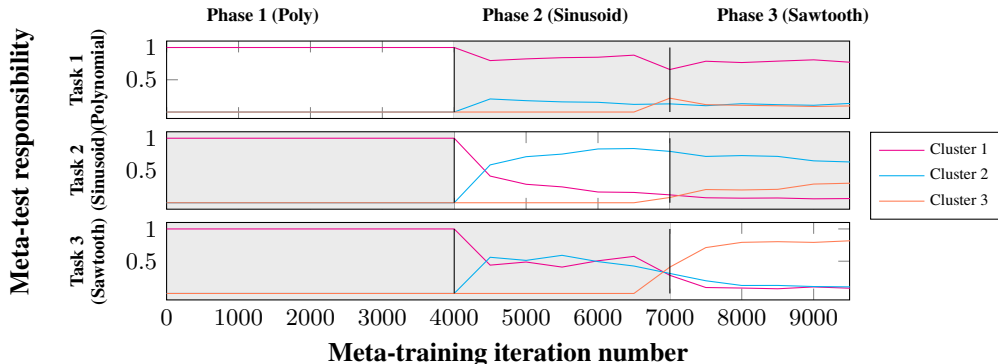

**Figure 7:** Task-specific per-cluster meta-test responsibilities $\gamma^{(\ell)}$ for both active and unspawned clusters. Higher responsibility implies greater specialization of a particular cluster (color) to a particular task distribution (row).

It is clear from Figure 8 that neither of our algorithms suffer from catastrophic forgetting to the same degree as the parametric baselines. In fact, at the end of training, both of our methods outperform all the parametric baselines on the first and second task.

**Results: Specialization.** Given the higher capacity of the parametric baselines and the inherent degree of similarity between the filtered *mini*ImageNet task distributions (unlike the regression tasks in the previous section), the parametric baselines perform better on each task distribution while during its active phase. However, they quickly suffer from degradation once the task distribution shifts. Our approach does not suffer from this phenomenon and can handle non-stationarity owing to the credit assignment of a single task distribution to a specialized cluster. This specialization is illustrated in Figure 9, where we track the evolution of the average cluster responsibilities on the meta-test dataset from each of the three *mini*ImageNet few-shot classification tasks. Each cluster is specialized so as to acquire the majority of a single task distribution's test set assignments, despite the degree of similarity between tasks originating from the same source (*mini*ImageNet). We observed this difficulty with the non-monotone improvement of parametric clustering as a function of components in Section 4.

## 7 Related Work

**Meta-learning.** In this work, we show how changes to the hierarchical Bayesian model assumed in meta-learning [21, Fig. 1(a)] can be realized as changes to a meta-learning algorithm. In contrast, follow-up approaches to improving the performance of meta-learning algorithms [*e.g.,* 30, 15, 20] do not change the underlying probabilistic model; what differs is the inference procedure to infer values of the global (shared across tasks) and local (task-specific) parameters; for example, [20] consider feedforward conditioning while [15] employ variational inference. Due to consolidation into one set of global parameters shared uniformly across tasks, none of these methods inherently accommodate heterogeneity or non-stationarity.

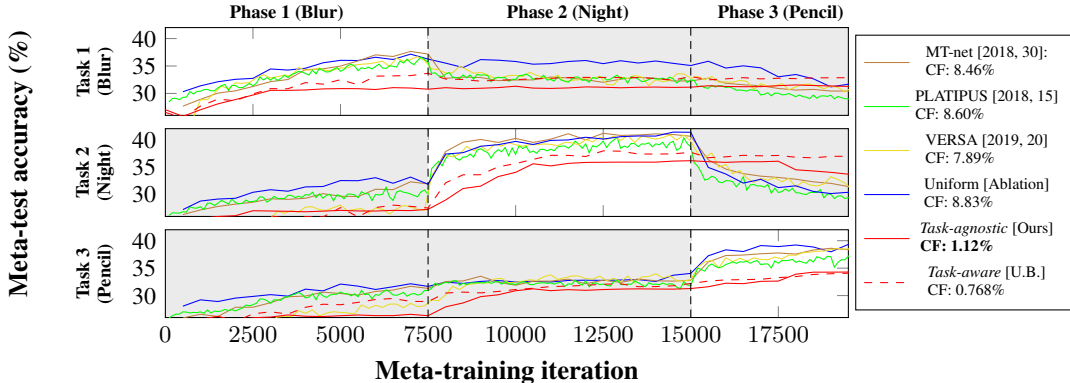

**Figure 8:** Results on the evolving dataset of filtered *mini*ImageNet few-shot classification tasks (higher is better). Each panel (row) presents, for a specific task type (filter), the average meta-test set accuracy over cumulative number of few-shot episodes. We additionally report the degree of loss in backward transfer (catastrophic forgetting, **CF**) in the legend. This is calculated for each method as the average drop in accuracy on the first two tasks at the end of training (lower is better; U.B.: upper bound).

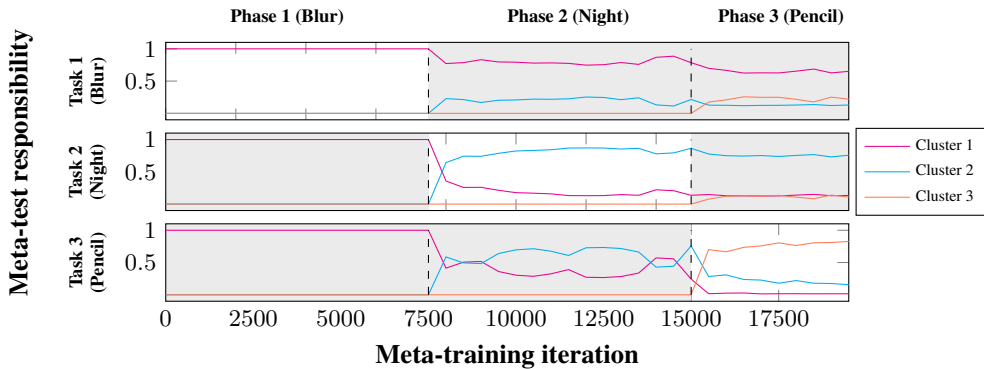

**Figure 9:** Task-specific per-cluster meta-test responsibilities $\gamma^{(\ell)}$ for both active and unspawned clusters. Higher responsibility implies greater specialization of a particular cluster (color) to a particular task distribution (row).

**Continual learning.** Techniques developed to address the catastrophic forgetting problem in continual learning, such as elastic weight consolidation (EWC) [28], synaptic intelligence (SI) [58], variational continual learning (VCL) [38], and online Laplace approximation [42] require access to an explicit delineation between tasks that acts as a catalyst to grow model size, which we refer to as *task-aware*. In contrast, our non-parametric algorithm tackles the *task-agnostic* setting in which the meta-learner recognizes a latent shift in the task distribution and adapts accordingly.

# 8 Conclusion

Meta-learning is a source of learned inductive bias. Occasionally, this inductive bias is harmful because the experience gained from solving a task does not transfer. Here, we present an approach that allows a probabilistic meta-learner to explicitly modulate the amount of transfer between tasks, as well as to adapt its parameter dimensionality when the underlying task distribution evolves. We formulate this as probabilistic inference in a mixture model that defines a clustering of task-specific parameters. To ensure scalability, we make use of the recent connection between gradient-based meta-learning and hierarchical Bayes [21] to perform approximate *maximum a posteriori* (MAP) inference in both a finite and an infinite mixture model. Our work is a first step towards more realistic settings of diverse task distributions, and crucially, *task-agnostic* continual learning. The approach stands to benefit from orthogonal improvements in posterior inference beyond MAP estimation (*e.g.,* variational inference [27], Laplace approximation [32], or stochastic gradient Markov chain Monte Carlo [33]) as well as scaling up the neural network architecture.

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
