[Supplementary Material · supp.pdf]

# A   Extended related work

**Multi-task learning.**   [44] demonstrated that negative transfer can worsen generalization performance, and avoidance of negative transfer has motivated much work on hierarchical Bayes in transfer learning and domain adaptation [*e.g.,* 29, 57, 16, 11, 54]. Closest to our proposed approach is early work on hierarchical Bayesian multi-task learning with neural networks that places a prior only on the output layer [24, 2, 46, 50]. In contrast, we place a non-parametric prior on the full set of neural network weights. Furthermore, none of these approaches were applied to the episodic training setting of meta-learning. Similar to our point estimation procedure, [24] and [50] propose training a mixture model over the output layer weights of a neural network using MAP inference. However, these approaches do not scale well to all the layers in a network as performing full passes on the dataset for inference of the full set of weights is computationally intractable in general.

**Clustering.**   Incremental or stochastic clustering was considered in the EM setting in [37]. and in the $K$-means setting in [48]. [31] conducted online learning of a non-parametric mixture model using sequential variational inference. A key distinction between our work and these approaches is that we leverage the connection between empirical Bayes in a hierarchical model and gradient-based meta-learning [21] to use a MAML-like [14] objective as a log posterior surrogate. This allows our algorithm to make use of a scalable stochastic gradient descent optimizer instead of alternating a maximization step with an inference pass over the full dataset [*c.f.,* 50, 3].

Our approach is also distinct from recent work on gradient-based clustering [22] since we employ the episodic batching of [53]. This can be a challenging setting for a clustering algorithm, as the assignments need to be computed using, for example, $K = 1$ examples per class in the 1-shot setting.

**Contrasting the batch and stochastic settings.**   In the stochastic setting, access to past data is unavailable, and so none of the standard algorithms and heuristics for inference in non-parametric models are applicable [*e.g.,* 26, 25]. In particular, our proposed algorithm does not refine the cluster assignments of previously observed points by way of multiple expensive passes over the whole dataset.

In contrast, we incrementally infer model parameters and add components during episodic training based on noisy estimates of the gradients of the marginal log-likelihood. Moreover, we avoid the need to preserve task assignments, which is potentially harmful due to stale parameter values, since the task assignments in our framework are meant to be easily reconstructed on-the-fly using the E–STEP with updated parameters $\boldsymbol{\theta}^{(0)}, \ldots, \boldsymbol{\theta}^{(L)}, G$.

*Maximum a posteriori* **estimation as iterated conditional modes.**   Due to the high-dimensionality of the parameter set of neural networks, we consider a mode estimation procedure based on iterated conditional modes (ICM) [6, 59, 55, 41] that can leverage gradient computation instead of the expensive process of Gibbs sampling. iterated conditional modes (ICM) is a greedy strategy that iteratively maximizes the full conditional distribution for each variable (*i.e.,* computes the MAP estimate), instead of sampling from the conditional as is done in Gibbs sampling [55]. This leads to a fast point-estimation of the DPMM parameters in which we only need to track the means of the cluster priors.

**Alternative inference procedures in probabilistic mixtures.**   A standard approach for estimation in latent variable models, such as probabilistic mixtures, is to represent the distribution using samples produced via some sampling algorithm.The most widely used is the Gibbs sampler [35, 17], which draws from the conditional distribution of each latent variable, given the others, until convergence to the posterior distribution over all the latents. However, in the setting of latent variables defined over high-dimensional parameter spaces such as those of neural network models, using a sampling algorithm such as the Gibbs sampler is prohibitively expensive [36, 34]. Instead of sampling, one can fit factorized variational distributions to the exact distribution $p(\phi, z|x) \approx q(\phi)q(z)$ [18, 7]. It should be noted that we do not claim that our method of point estimation in the DPMM is the most accurate method for posterior inference but we leave improved approximate inference extensions to future work.

The main drawback of using point estimates for a non-parametric mixture estimation is the inability to leverage the diffusion of the global prior $G_0$ when computing the likelihood of a new cluster. Highly concentrated parameter estimates for non-empty clusters should lead to low likelihoods for outlier tasks, whereas the diffused global prior should be better at capturing a wider variety of tasks.

Nonetheless, point estimation is a necessary trade-off between computation and accuracy. To allow for a more accurate estimate of the likelihood, we experimented with simulating a normal centered at the global prior mean with a variance hyperparameter that can be annealed over time to account for increased certainty about the prior choice. We can then compare the average cluster responsibility to the threshold. Another interesting extension we experimented with was to compute the gradient for each of the samples and average over the number of samples as to approximate the expectation of the gradient under the global prior. However, we found this to be less stable than simply comparing the cluster responsibilities to the threshold.

## B   *Maximum a posteriori* estimation in the Dirichlet process mixture model

From (4) and using a conditional mode estimate for task-specific parameters $\boldsymbol{\phi}_j$,

$$
\log p\left(\boldsymbol{z}_j = \ell \mid \boldsymbol{x}_{j_{1:M}}, \boldsymbol{z}_{1:j-1}, \boldsymbol{\theta}^{(\ell)}\right) \approx \begin{cases} \log n^{(\ell)} + \log p(\boldsymbol{x}_{j_{1:M}}|\hat{\boldsymbol{\phi}}_j^{(\ell)}) + & \\ \log p(\hat{\boldsymbol{\phi}}_j^{(\ell)}|\boldsymbol{\theta}^{(\ell)}) & \text{for } \ell \leq L \\ \log \zeta \ + \ \log p(\boldsymbol{x}_{j_{1:M}}|\hat{\boldsymbol{\phi}}_j^{(\ell)}) \ + & \\ \log(\hat{\boldsymbol{\phi}}_j^{(\ell)}|\boldsymbol{\theta}^{(0)}) & \text{for } \ell = L + 1 \,. \end{cases}
$$

$$(5)$$

## C   Experimental setup

### C.1   Dataset details

**Few-shot regression**

- Polynomial wave (Figure 4a):
$$y = \sum_i a_i x^{p_i}$$
  and $a \sim \mathcal{U}(-5.0, 5.0)$.
- Sinusoid wave (Figure 4b):
$$y = a \sin(x - \phi)$$
  where $\phi \sim \mathcal{U}(0, \pi)$ and $a \sim \mathcal{U}(0.1, 5.0)$.
- Sawtooth wave (Figure 4c):
$$y = -\frac{2a}{\pi} \arctan(\cot(\frac{x\pi}{\phi}))$$
  where $\phi \sim \mathcal{U}(0, \pi)$, $a \sim \mathcal{U}(0.1, 5.0)$.

### C.2   Hyperparameter choices

#### C.2.1   *Mini*ImageNet few-shot classification.

We use the same data split, neural network architecture, and hyperparameter values as in [14] for common components. We use $\tau = 1$ for the softmax temperature and the same initialization as [14] for the global prior $G_0$. We determine an iteration number for early stopping using the validation set.

#### C.2.2   Continual few-shot regression.

Our architecture is a feedforward neural network with 2 hidden layers with ReLU nonlinearities, each of size 40. We use a meta-batch size of 10 tasks (both for the inner updates and the meta-gradient updates) for 5-shot regression. Our non-parametric algorithm starts with a single cluster ($L_0 = 1$ in Algorithm 3). In these experiments, we set the spawning threshold $\epsilon = 0.95T/(L + 1)$, with $L$ the number of non-empty clusters and $T$ the size of the meta-batch. We use the mean-squared error for each task as the inner loop and meta-level objectives.

### C.2.3 Continual few-shot *mini*ImageNet classification.

We use the same data split, neural network architecture, and hyperparameter values as in [14] for common components. We use a meta-batch size of $4$ tasks, start with a single cluster, and set the spawning threshold to the same formula as in Section C.2.2. We use the multi-class cross entropy error for each task as the inner loop and meta-level objectives. More details on the the practical implementation for image datasets of the non-parametric algorithm can found in Section D.

# D  Practical and implementational details

## D.1  *Task-aware vs. task-agnostic*

Since a cluster is not well-tuned immediately after its creation, we consider a cool-down period after the spawning of each new cluster where we do not consider the creation of new clusters for a fixed number of iterations, and we freeze the updating of existing clusters for a same number of iterations. This allows the newly-created cluster to take enough gradient updates in order to move from its global prior initialization, allowing it to sufficiently differentiate from the global prior.

This experimental paradigm also allows us to approximate the *task-aware* algorithms of prior work [*e.g.,* 28, 58, 38, 42] which require access to an explicit delineation between tasks that acts as a catalyst to grow model size. For the *task-aware* non-parametric mixture results reported in the experiments, we set this cool-down period to be exactly the length of the training phase for the appropriate dataset; therefore, clusters which are not meant to be specialized for the active dataset are not updated. In contrast, the *task-aware* results consider a cool-down period of $1k$ iterations, which is less than 15% of the active period for each dataset. Extensions to this fixed cool-down period could consider the rate of learning in the active cluster in order to detect when the new component has been sufficiently fit to the new task.

## D.2  Practical extensions to the non-parametric algorithm

The penalty term of $\log n^{(\ell)}$ or $\log \zeta$ is necessary to regularize the likelihood of a potential new cluster in order to limit overspawning. However, in the setting where the likelihood is approximated by the loss function of a complex neural network, as in the case for most meta-learning applications, there is a large difference in orders of magnitude between the loss value (especially for the cross-entropy function) and the penalty term, even after a single batch of assignments. Furthermore, the classical log observation count $\log n$ term is misaligned with our stochastic setting for two reasons. First, since we do not re-evaluate over the whole dataset for every meta-learning episode, we are thus more concerned with the relative number of task assignments over recent iterations than the total number of assignments over the duration of training. Second, the number of tasks to be assigned can grow too large in the stochastic setting (e.g. $60k$ for *mini*ImageNet) which exacerbates the already large difference in orders of magnitudes between the loss function and the penalty term.

Accordingly, we propose two changes; First, we compute the observation based on a moving window of fixed size (5 in the experiments). Second, we apply a coefficient, which can be tuned, to the log observation count in (4). This provides more flexibility to our meta-learner as it allows it to apply to any black-box function approximator which might exhibit losses of orders of magnitudes smaller than those expected of classical probabilistic models. While the moving window size and CRP penalty coefficient terms are somewhat interdependent, we propose them as a simple starting point to tune this non-parametric meta-learner beyond what is empirically explored in this paper.

Note that without such changes in the stochastic setting of meta-learning, a nonparametric algorithm would be unable to spawn a new cluster after the first handful of iterations. Even if we were to lower the threshold $\epsilon$, multiple almost identical clusters would be spawned in the first few iterations before it would be impossible to spawn anymore. Furthermore, the clusters would be nearly identical given the small step size of a gradient update for each meta-learning episode. Finally, this would be computationally intensive since unlike the typical applications of non-parametric mixture learning where one can afford to spawn hundreds of components then prune them over the training procedure.

### D.3 Thresholding

A marked difference that is not immediate from the Gibbs conditionals is the use of a threshold on the cluster responsibilities, detailed in the `E-STEP` in Subroutine 4, to account for noise from stochastic optimization when spawning a cluster on the basis of a single batch. This threshold is necessary for the stochastic mode estimation procedure of Algorithm 3, as it ensures that a new cluster's responsibility needs to exceed a certain value before being permanently added to the set of components.

If a cluster has close to an equal share of responsibilities as compared to existing clusters after accounting for the CRP penalty $\log n^{(\ell)}$ or $\log \zeta$, it is spawned. Accordingly, this approximate inference routine still preserves the preferential attachment ("rich-get-richer") dynamics of Bayesian nonparametrics [41]. A sequential approximation for non-parametric mixtures with a similar threshold was proposed in [31] and [51], in which variational Bayes was used instead of point estimation in a DPMM.

### D.4 Pruning heuristics

None of the results reported in our experiments used a pruning heuristic as we used a rather conservative hyper parameter setting that deters overspawning. We did however explore different heuristics which could work in more general settings, especially in the presence of many more latent clusters of tasks than considered in the experimental settings in this work. One such heuristic is to prune small clusters that have received disproportionately few assignments over a certain number of past iterations. Another is to evaluate the functional similarity of two clusters by computing an odds-ratio statistic for the assignment probabilities to each cluster over a set of validation tasks. If the odds-ratio statistic is below a certain threshold, the smaller cluster can be pruned.

### D.5 Estimating the CRP hyperparameters

We fixed $\alpha$ at the size of the meta-batch. An alternative is to place a $\Gamma(1, 1)$ on the concentration parameter. Based on the likelihood , the posterior is then proportional to $p(\alpha|N, K) \propto \frac{\Gamma(\alpha)}{\Gamma(\alpha+N)} \alpha^K e^{-\alpha}$ This is not a standard distribution but [39] have shown that $\log p(\alpha|N, K)$ is log-concave and methods such as L-BFGS have been used successfully in prior works. Alternatively, if we have some prior knowledge about the expected number of clusters, we can compute $\alpha$ based on $E[K] = \alpha \log N$. For the window-size, we considered an initial size of 20 iterations that can grow as more cluster are considered.

### D.6 Implementation details

We implemented both of our parametric and non-parametric meta-learners in TensorFlow (TF) [1]. We considered 2 different settings for the `M-STEP` optimization:

- Train each cluster's parameters separately based on its corresponding loss function in an alternating manner closest to the classic EM algorithm.

- Train all cluster weights simultaneously using a surrogate loss over all validation batches.

Since the latter better leverages the differentiability of softmax-clustering and performed better empirically, we used it to report all experimental results.

### D.6.1 Nonparametric Implementation

For the nonparametric algorithm, we chose the first approach to the `M-STEP` by constructing separate optimizers for each cluster's parameters. We pre-allocate a set of weights and use a mask during training to discard the parameters of empty clusters due to the static nature of TF graphs. When the algorithm exhausts the set of pre-allocated weights, we simply construct more network weight and reinitialize our optimizers.

### D.6.2 CRP global prior

The likelihood of a new cluster is sensitive to the choice of a base measure or prior prior, $G_0$ on the cluster hyperparameters. Our gradient-based point estimation does not make any modeling assumption on the distribution of the weights, rendering the problem of principally updating the base measure, after or during training, non-trivial. We chose to initialize all weights with zero-mean normals in the fully-connected layers. For the convolutional layers, we leveraged Xavier initialization [19] similarly to prior work [14] in meta-learning.

However, such initialization is poor in the non-parametric for most non-trivial regression or classification tasks. Therefore, in the nonparametric setting, we start with a single cluster for a fixed number of iterations. We then initialize all clusters with the weights of the first clusters. This set of weights can be considered as the mean of the base measure or global prior in our setting.

We periodically update the global prior using a uniform average of the parameters of the existing clusters. This can be done by simply averaging over the parameter of the non-empty clusters as weighted by their sizes. Note that, we found that performing weighted KDE smoothing with a small bandwidth hyperparameter to perform slightly better than the average which is to be expected for neural network parameters. The number of iterations between updates of the global prior is a hyperparameter that we tune on the validation set. It is also possible to continuously, but less frequently over time, update this global prior as more data is encountered.