[Reviews · NeurIPS 2019]

Reviewer 1



Originality: I really liked the idea of modeling MAML-like approaches using nonparametric Bayesian priors and I am not familiar with other work that does that. Thus, I consider the proposed method novel. More specifically, I consider it a novel combination of existing methods, that exploits an interesting connection between them. Quality: The paper is of good quality both in terms of key contributions and in terms of how the contributions. are presented. Clarity: The paper is well-written and organized, and very easy to follow, given a bit of background in Bayesian methods. Significance: As mentioned in my earlier comments, I consider this work significant. One other comment is that I really like the extensive discussion of related work, both in the paper and in the supplementary material. I always find that very useful and often ignored so I was positively surprised in your case.

Reviewer 2



Overall, this is a very strong submission. It is well-written, timely, clear, and includes several significant and novel contributions. The technical contributions are well-developed, and it does a very nice job addressing a challenging problem in meta-learning. Although the novelty is high, it is related to several papers on online MAML and MAML with task-clustering that appeared in ICML 2019. Although I know the timing is very tight between these, so it isn't reasonable to expect empirical comparison, those works should at least be cited and compared qualitatively in related work. I would like to see more in the paper addressing the tension between the ability to fit the meta-parameters to each task clusters and the ability to generalize. This has a nice interpretation in the Bayesian setting that could be called out further. The clarity of Section 3 would be improved by adding a plate diagram or illustrative figure showing the relationship among the variables. POST-RESPONSE Authors -- thanks for your clear and engaging response. Please do add the plate diagram back in, at least as supplemental material, but preferably in the main paper.

Reviewer 3



The problem is interesting. The combination of continual learning and meta-learning is novel. The method is technically sound. The experiments on toy tasks are well-designed. The paper is clearly written. ======= UPDATE ======= I have read the authors' response and understood the difficulty of finding a naturalistic dataset.

[Author Response · NeurIPS 2019]

We thank the reviewers for their constructive comments and share their enthusiasm about combining techniques
from non-parametric Bayes with neural networks in order to tackle problems such as negative transfer and adaptive
complexity. We briefly state our contributions in contrast to prior work (p.w.) in meta-, continual, and online learning:

• p.w. in *meta-learning* does not avoid catastrophic forgetting due to a shifting task distribution (the "online" setting);

• p.w. in *continual learning* requires an explicit delineation between tasks as a catalyst to adapt model size [4, 5, 9] and
does not present a benchmark for few-shot learning (i.e., episodic batching as in [7]); and

• p.w. in *online learning* does not measure negative backward transfer ("catastrophic forgetting"), since all task types
remain active under the training distribution upon their introduction [1, 8].

**R1: Figure 8.** We note that all baselines suffer a greater degree of catastrophic forgetting than our method (as measured
by the average decrease in inactive task performance from its value at the end of its active phase; baselines: $\approx 8\%$ vs.
ours: $\approx 1\%$) despite the fact that our method has restricted capacity (starts with one component) at the beginning of
training. The peaked assignment distributions in Figure 9 evidence that the reduction in catastrophic forgetting is due to
incremental learning of specialized clusters. We welcome the reviewer's feedback on how to improve the presentation.

**R1: "In Figure 6, the blue line is not visible..."** Note that during Phase 1, Tasks 2 and 3 are inactive, thus leading to
the exploding losses of the uniform mixture baseline (corresponding to the blue line in Figure 6), as the uniformity
constraint does not allow the mixture to selectively activate a single component to be trained for the single active task.
The exact degree to which the loss explodes (i.e., what is truncated in the first panel in Figure 6) is not informative
since the comparisons (the other ablation and the full method) do not exhibit any such increase (i.e., their relative
improvement is infinite). Thus we did not expand (or log-transform) the $y$-axis in the third row of Figure 6, as doing so
would obscure behavior in Phases 2 and 3, which is of primary interest.

**R1: "Margins are often reduced too visibly..."** We apologize for the small vertical spacing around the equations
which might have created visual clutter; we will fix this for the camera-ready copy by trimming the writing.

**R2: Qualitative comparison to "online MAML."** We believe the reviewer is referring to [1]. We note that the online
setting is quite different from the continual learning setting, even though both assume a non-stationary data distribution.
In [1], all previous data is available and, as such, there is no issue of negative backward transfer ("...we sidestep
the problem of catastrophic forgetting by maintaining a buffer of all the observed data" [pg. 4 of 1]). The focus in
[1] is instead on improving positive forward transfer; in contrast, we explicitly address negative backward transfer
(catastrophic forgetting) by adding model components via a non-parametric prior.

**R2: Qualitative comparison to "MAML with task clustering."** We believe the reviewer is referring to [8]. The
results in Figures 4 and 7 of [8] result from experimental setups in which new image datasets (Bird, Texture, Air-
craft, Fungi) are incrementally added to the training pool. This is a subtle point that is not identified in the paper,
but is evident after inspecting the code repository; see `https://github.com/huaxiuyao/HSML_Dynamic/blob/`
`1af8e8068676df589a5e95b787190eda729c8a8a/data_generator.py#L230-L242` for the implementation, and
note that a training batch is sampled from any dataset up to and including the most recently added. While this results
in non-stationarity, each dataset type is active from the time it is introduced until training is terminated. Analogously
to [1], this prevents catastrophic forgetting; thus [8] does not address the general setting of continual learning.

**R2: "'...tension between the ability to fit the meta-parameters to each task clusters and the ability to generalize."**
Playing around with extrapolation under a Bayesian lens is a great suggestion. We would welcome any specific
recommendations from the reviewer on this point, and we will certainly look into it to enrich this work.

**R2: "...clarity of Section 3 would be improved by adding a plate diagram or illustrative figure."** We removed a
plate diagram due to space constraints, but will make sure to include it in the appendix in the camera-ready copy.

**R3: On more naturalistic data.** We were unsuccessful in finding an open-sourced, naturalistic dataset with a
standardized few-shot episodic batching as described in [7]; without this, the results would be difficult to understand in
reference to prior work in meta-learning. Moreover, most of the recent research on distributional shift and perturbation
analysis in computer vision relies on modifications of ImageNet similar to our modification of *mini*ImageNet [e.g.,
3, 6, 2]. We opted for transformations to common datasets to better understand the behavior of our method in a variety
of settings where meta-learning is known to perform adequately (i.e., standardized few-shot regression and few-shot
classification). We hope our work will inspire the creation of datasets that present naturalistic non-stationarities along
49/50 the lines of what the reviewer suggests.

[1]  C. Finn, A. Rajeswaran, S. Kakade, and S. Levine. Online meta-learning. In *ICML*, 2019.
[2]  R. Geirhos et al. ImageNet-trained CNNs are biased towards texture; increasing shape bias improves accuracy and robustness. In *ICLR*, 2019.
[3]  D. Hendrycks and T. Dietterich. Benchmarking neural network robustness to common corruptions and perturbations. In *ICLR*, 2019.
[4]  J. Kirkpatrick et al. Overcoming catastrophic forgetting in neural networks. *Proceedings of the national academy of sciences*, 114(13):3521–3526, 2017.
[5]  C. V. Nguyen, Y. Li, T. D. Bui, and R. E. Turner. Variational continual learning. In *ICLR*, 2017.
[6]  S. Rabanser, S. Günnemann, and Z. C. Lipton. Failing loudly: An empirical study of methods for detecting dataset shift. *arXiv preprint arXiv:1810.11953*, 2018.
[7]  O. Vinyals, C. Blundell, T. Lillicrap, D. Wierstra, et al. Matching networks for one shot learning. In *Advances in neural information processing systems*, pages 3630–3638, 2016.
[8]  H. Yao, Y. Wei, J. Huang, and Z. Li. Hierarchically structured meta-learning. In *ICML*, 2019.


[Meta-Review · NeurIPS 2019]

The reviewers all agreed this is a solid contribution and well-presented. The topic is timely and the reviewers appreciate the extensive discussion to related work (like MAML), both in the paper and in the supplementary material. The authors also praised how the technical contributions are developed.